# Interactive Educational Toy Design Strategies for Promoting Young Children’s Garbage-Sorting Behavior and Awareness

**DOI:** 10.3390/ijerph20054460

**Published:** 2023-03-02

**Authors:** Zhenwei You, Tingting Yang, Zhe Li, Yi Li, Ming Zhong

**Affiliations:** Design Department, School of Digital Media and Design Arts, Beijing University of Posts and Telecommunications, Beijing 100876, China

**Keywords:** educational toys, garbage classification, interaction design

## Abstract

Existing educational toys for teaching garbage classification fail to teach about its benefits and positive results. Thus, children do not fully understand the logic behind garbage classification. We summarized the design strategies of garbage classification educational toys according to parents’ evaluations of existing toys and the literature on children’s memory characteristics. Presenting children with all the system information related to garbage classification is essential for their logical understanding. Using interactive formats and personified images enhances children’s desire to play with toys. Based on the above strategies, we designed an intelligent trash can system toy: Incorrect garbage input displays an uncomfortable expression and sad voice. Correct garbage input triggers happy expressions and positive sounds. An animated story then shows how the garbage is treated and recycled into something new. The results of a contrast experiment showed that the accuracy rate of children’s garbage classification was significantly raised after playing with the designed toy for two weeks. The toy also promoted children’s garbage-sorting behavior in daily life. When seeing trash misclassified, the children would correct the mistakes and take the initiative to share relevant knowledge about garbage disposal.

## 1. Introduction

China’s garbage classification has started late [1] but has drawn on the advanced methods of Japan, Germany, and other countries that have taken the lead in garbage classification practices. However, the effect of China’s initiative is not ideal due to the public’s relatively weak awareness of environmental protection [2,3,4]. The existing improvement measures include increasing garbage classification propaganda, improving the long-term law enforcement mechanism, and strengthening the control of waste reduction at the source. Building a beautiful and environmentally friendly country is difficult when using only slogans and mandatory punishment, relying on garbage instructor supervision and mandatory delivery at fixed times, and failing to conduct environmental education and develop awareness to incorporate garbage classification into everyday lifestyle [5,6,7].

A person is more likely to develop long-term, continuous, and stable habits in childhood [8]. The preschool age is a critical stage for habit formation [9]. Children aged 3–6 years are at the initial stage of independent exploration and formation of independent consciousness and behavior [10]. A child’s positive attitude and behavioral tendencies in the process of activities are valuable qualities necessary for lifelong learning and development [11]. Therefore, it is necessary to incorporate garbage classification lessons into early childhood education to teach effective and sustainable habits [12].

The common educational products for preschool children’s domestic waste classification can be divided into two categories: popular science books and educational toys [13,14]. Rote memory and intensive training do not conform to preschool children’s learning characteristics and receptivity. Therefore, the traditional model of environmental education for children often suffers from ineffectiveness and a lack of environmental action [15]. Because preschool children are in a special stage of reading pictures, popular science books strive to create interesting and appropriate reading materials to enhance parents’ scientific ability and teachers’ professional quality [11]. However, educational toys are useful for more than just entertainment and play; they create play affordances for children [16,17]. As children interact with toys, they can gradually perceive and recognize these affordances, which can help them learn new skills, develop into lifelong learners [18,19], and encourage the exploration of their interests later in life [20,21]. Thus, a continuous education process should be established when using garbage classification toys with young children [22]. The aim is to help children grow accustomed to and master the complex operation of garbage classification. The results would not only make garbage classification a matter of public awareness, but also common sense [23].

Previous studies drew on the design methods and concepts of educational games and put forward the corresponding design strategies of garbage classification educational toys. Castellano et al. proposed the design strategy of introducing a competition mechanism. They used a social robot to compete with children in garbage identification and classified garbage disposal to improve the effect of garbage classification education [23]. Jiang and Xu proposed strengthening the memory of garbage classification knowledge and garbage classification ability by making children play the role of stories when using toys. These design strategies have a certain universality, but they are not targeted [24].

The purpose of this study is to explore specific educational toy design strategies for promoting young children’s garbage-sorting behavior and awareness. First, through the analysis of the existing preschool garbage-sorting toy products and methods to improve children’s memory of garbage classification knowledge, we summarized the design strategies of preschool garbage-sorting toys. Then, we designed an intelligent trash can system toy to test and verify these design strategies.

## 2. Analysis of Preschool Garbage-Sorting Toy Products

The children’s toy industry has been showing a trend of rapid development, aligning with the rapid development of the manufacturing industry, the Internet, and the integrated circuit industry [25]. As a new toy type, garbage-sorting toys are rarely seen in China’s market. Therefore, learning from and drawing on the advantages of other preschool educational toys to explore feasible design strategies for domestic garbage-sorting educational toys is essential.

### 2.1. Comparison of Preschool Garbage Classification Educational Toys and Preschool Educational Toys

We adopted an online comparative research method, choosing the e-commerce platform Taobao as the source. Taobao, founded by Alibaba Group, is a large online retail and business district in the Asia-Pacific region. Taobao is very popular in China and has nearly 500 million registered users [26]. The top 50 best-selling products of *preschool garbage classification educational toys* and the top 50 best-selling products of *preschool educational toys* in China were summarized and analyzed from the aspects of type and function.

Table 1 shows that card games and transportation-type toys occupy 52% and 44% of the garbage classification educational toy market, respectively. These toys have two main functions: promoting correct garbage classification and understanding the garbage truck workflow. Additional types of preschool educational toys include Legos, chess, card games, tool-type toys, life simulation toys, and remote-control toys. These products also fulfill more functions, e.g., knowledge learning, skill learning, and intellectual development. However, garbage classification educational toys only focus on one knowledge point: correct garbage classification. Preschool educational toys typically focus on systematic knowledge and skill development. For example, children playing with Legos must first recognize the shape of each unit, then understand the various connection methods between units, and, finally, think about how to build their desired object. This type of play requires a continuous thinking process and constitutes a knowledge system. Thus, garbage classification educational toys should also constitute a knowledge system to stimulate continuous thinking.

### 2.2. Parents’ Evaluation and Expectation of Existing Preschool Garbage Classification Educational Toys

In order to evaluate the impacts of existing preschool garbage classification educational toys, we distributed the two best-selling toys to twenty children (ten males and ten females) aged 3–6 years (M = 4.75, SD = 0.89) from a kindergarten in Beijing: one card game and one garbage truck toy (see Figure 1). After two weeks of use, we conducted a focus-group interview to evaluate their use effect and expectations according to the parents of these twenty children.

The children’s parents reached a general consensus regarding the toys’ use effects. Card games had an effect on garbage-sorting education, but children typically did not enjoy them. The process of using card games is similar to memorizing words (rote memory training), which children find boring. Garbage truck toys were more popular with the children. However, when using them, the children focused on playing with the toy vehicles rather than on sorting garbage. Notably, when sorting garbage, the children often inquired why it was classified as such and what was the effect of classification.

We have summarized all the statements made by the parents in the interview. The high-frequency keywords were analyzed regarding expectations and according to the summarized interview documents. Among them, the keyword “explanation” appeared 28 times, “interaction” appeared 24 times, “interesting” appeared 23 times, and “experience” appeared 18 times. “Explanation” refers to the meaning, benefits, and positive effects of garbage classification. “Interaction” refers to children interacting with toys and the toys providing children with timely information feedback. For example, when garbage is put into the wrong bucket, the toys should immediately notify the children. “Interesting” refers to toys being interesting and attracting children’s attention to enhance their willingness to learn. “Experience” refers to children practicing the skills and solving problems themselves.

Therefore, effective garbage classification educational toys should be attractive and engaging rather than rely on rote memorization. Additionally, they should provide children with an interesting and interactive experience when presenting the garbage classification process.

## 3. How to Improve Children’s Memory of Garbage Classification Knowledge

Barreto et al. argued that environmental educational toys should be designed according to children’s cognitive development and their learning skills [27]. The first task of garbage classification education is to improve children’s memory of garbage classification knowledge. Memory refers to the brain’s process of remembering, maintaining, recalling, and recognizing things it has experienced. The brain stores cognition of the world in the form of information. Thus, information storage is a memory process [28]. Memory is a basic ability built on senses and perception. It develops gradually with the accumulation of experiences, and is the foundation of high-level cognitive activities, such as thinking and imagining [29]. Through the process of repeated memories, children form their own experience systems. Thus, school-aged children gradually regard the brain as an active and constructive unit [30]. It stores not only a copy of the objective reality but also an explanation of reality. Children’s memory characteristics mainly comprise the following:A significant improvement in memory ability, including memory span, accuracy, and memory speed, e.g., the ability to store speech, store visuospatial information, control attention, and extract information from long-term memory—which increases with age [31].Meaningful memory (meaningful association) proves more effective than rote memory (meaningless association). Children’s understanding ability is enhanced with the improvement of thinking and speech abilities. Additionally, meaning memorization gradually plays a leading role in rote memorization.The memory of concrete images is stronger than that of abstract words. Children often learn and memorize abstract words with the help of specific images, and the memory of specific images still plays a major role in memory abilities [32].Emotion affects memory’s effect. Children tend to recall familiar things and emotional events. The memory performance of positive emotional words is better than that of negative emotional words and neutral words [33].

Presenting all the systematic information related to garbage classification is necessary to strengthen the memory of garbage classification and arouse children’s meaningful associations, e.g., how garbage is treated after classification and the resulting effects. By directly linking the results with the classification behavior, children can better understand the necessity of garbage classification. In addition, materials should convey information via images (minimizing text information) and create a positive emotional atmosphere for children.

## 4. Summary of Design Strategies

According to the results of the analysis of preschool garbage-sorting toy products and methods to improve children’s memory of garbage classification knowledge, we summarized the design strategies of preschool garbage-sorting toys into four levels: educational content, use, form, and atmosphere (Table 2). The educational content level should provide children with systematic knowledge, including garbage classification knowledge, how to deal with garbage, and the effects on garbage after treatment. In use, interaction should be developed to provide children with timely information feedback. The format should be interesting, attract children’s attention, and use more images than text. Finally, the toys should create a relaxed and pleasant atmosphere.

Interesting interactive formats and personified images (objects displaying human characteristics) are more attractive to children. These qualities enhance their desire to use a toy and result in stronger educational outcomes.

## 5. Design Practice and Effect Verification

### 5.1. Design Implementation

In China, garbage is divided into four categories and distinguished by differently colored trash cans: kitchen waste in green, recyclable waste in blue, harmful waste in red, and other waste in black. Based on the strategies we summarized, we designed an intelligent trash can system toy (see Figure 2).

Each can was mainly composed of a barrel body, a display, an object recognition sensor, and an audio system (see Figure 3). The recognition sensor was used to identify the garbage and to which category it belonged.

When users threw the garbage (toys) into the wrong trash can, it displayed an uncomfortable expression and used a sad voice. When users threw garbage (toys) into the correct trash can, it displayed happy expressions and made positive sounds (Figure 4a). The garbage thrown in the trash appeared in the form of cute cartoons on the monitor (Figure 4b).

An animated story showed how the garbage was treated and recycled into new items (Figure 5).

### 5.2. Participants

Twenty children (ten males and ten females) aged 3–6 years (M = 4.85, SD = 0.91) from a kindergarten in Beijing participated in the experiment to test the effects of the designed intelligent trash can system toy. These participants were re-recruited and there was no overlap between them and the participants of the evaluation experiment in Section 2.2. Before this experiment, the participants had never systematically learned about garbage classification. A text document was provided to students and parents, outlining the purpose and process of this study, and informed consent was obtained from all participants before the experiment.

### 5.3. Procedure

This experiment comprised three sessions: a pre-test, an intervention training using the designed toy, and a post-test. In the pre-test session, the children received a simple explanation of the different colored trash cans and their meanings. Then, they were asked to put 20 garbage toys (5 toys for each garbage type) into the corresponding trash cans. The delivery accuracy and the reaction duration were recorded.

After the pre-test, the children and their parents received instructions on how to use the intelligent trash can system toy we designed. In the subsequent two-week intervention training session, the parents were instructed to guide and encourage children to interact with the intelligent trash can system toy and play along with them for no fewer than 15 min per day.

During the post-session, the children’s garbage-sorting ability was tested again. The test procedure was consistent with the pre-test. After completing the post-test, the parents were required to attend a focus-group interview about the user experience of the intelligent trash can system toy. They were asked to evaluate the intelligent trash can system toy using a 5-point Likert scale (*very dissatisfied*, *not very satisfied*, *average*, *somewhat satisfied*, *very satisfied*) and discuss the phenomena and problems observed during the use process.

### 5.4. Results

#### 5.4.1. Accuracy of the Delivery

In order to determine whether children’s garbage classification accuracy improved after using the intelligent trash can system toy for two weeks, we conducted a t-test to compare the overall score of garbage placement and the scores of four types of garbage delivery between the pre-test and post-test. The delivery accuracy in the prediction test was verified using the Shapiro–Wilk test for normality. The variables’ data followed a normal distribution (*p* = 0.35).

The results showed that the overall average score *t*(38) = 5.62, *p* < 0.01, and the average scores of kitchen waste *t*(38) = 6.23, *p* < 0.01, harmful waste *t*(38) = 3.43, *p* < 0.01, recyclable waste *t*(38) = 4.98, *p* < 0.01, and other waste *t*(38) = 3.48, *p* < 0.01 were statistically significantly different before and after the intervention training (see Figure 6).

Through the two-week intervention training, the overall garbage classification accuracy of the twenty children increased by 31.25%. Among them, the correct classification rate of kitchen waste increased the most (40%), followed by recyclable waste (34%). The growth rate of harmful waste (25%) and other waste (26%) was relatively low.

#### 5.4.2. Reaction Duration

In order to comprehensively judge the improvement of children’s garbage classification ability, we further confirmed whether the reaction duration of children’s garbage classification changed after the intervention.

Through the Shapiro–Wilk test for normality, we verified the delivery accuracy in the prediction test. The data of variables followed a normal distribution (*p* = 0.89). The results showed that, although the overall mean reaction duration decreased by 650ms after the intervention, there was no significant difference between the pre-test and post-test *t*(38) = 1.61, *p* > 0.05. After the intervention implementation, the average reaction durations of the four types of garbage dropped by varying degrees. The average reaction durations of kitchen waste *t*(38) = 2.51, *p* < 0.05 and recyclable waste *t*(38) = 1.77, *p* > 0.05 in the pre-test and post-test showed significant differences. However, the average reaction durations of harmful waste *t*(38) = 1.45, *p* > 0.05 and other waste *t*(38) = 0.88, *p* > 0.05 in the pre-test and post-test showed no significant differences (see Figure 7).

#### 5.4.3. Parents’ Evaluations

Of the participating parents, 85% were very satisfied with the intelligent trash can system toy, and 15% were somewhat satisfied. The results of the focus-group interviews can be summarized into four points. First, the children were very interested in this toy and took the initiative to play with it. Second, the children began to classify garbage in their daily lives. When they saw other people misclassifying trash, they would correct their mistakes and take the initiative to share relevant knowledge about garbage disposal. Third, this toy promoted communication between parents and children. Children often discussed with their parents how a garbage item would be disposed of and what new things it would become after disposal. Fourth, some garbage classifications could be confusing, even for adults, especially the “other” type of waste.

## 6. Discussion

After playing with the intelligent trash can system toy for two weeks, the children’s ability to classify the four types of garbage improved to a certain extent. The accuracy of garbage classification was also significantly improved. Such results proved that our design strategy was appropriate and our design solution was effective. However, overall, the reaction speed of children’s garbage sorting did not improve significantly. In sorting kitchen waste, the reaction speed was significantly increased, while the average reaction durations of harmful waste, recyclable waste, and other waste showed no significant differences. This study’s subjects were preschool children aged 3–6 years. Some studies have shown that young children are more likely to accept empirically biased explanations [34,35]. Preschool children are familiar with kitchen waste, which is consistently present in their daily lives. It is visible and tangible, so it bears a more direct relationship to the real world and sensory counterparts, and is therefore more easily understood [36]. Their frequent contact and experience with this type of waste results in their faster classifying judgment. Relatively speaking, harmful waste and recyclable waste are abstract concepts for young children, and related information processing requires a process. Because children have limited experience with these types of waste and need to associate objects with concepts, the classification speed cannot be effectively improved. The other waste type requires a judgment based on excluding other waste types, which is a process problem. The process of judgment cannot be changed through intervention training, so the speed of judgment cannot be improved. In addition, sorting the other waste type might be an unreasonable expectation for young children. Even adults sometimes become confused and make incorrect judgments. However, the policies and regulations on waste classification can be further refined in the future.

The next step of research could be aimed at teenagers to explore and formulate design strategies suitable for improving their garbage classification skills. Researchers could also compare the differences in garbage classification cognitive learning between preschool children and adolescents.

In addition, gender may lead to different preferences for the same thing, which will affect the learning and use effects. In this study, there is no comparison between the effects of male children and female children using the designed toys. If there are differences, we could further explore gender-specific design strategies through experiments in future research to achieve better educational results.

## 7. Conclusions

This study is an exploration of intelligent interaction product design in the practice of garbage classification education for preschool children. Through the analysis of existing preschool garbage-sorting toy products and methods to improve children’s memory of garbage classification knowledge, we summarized the design strategies of garbage classification educational toys. In order to strengthen the educational effect of garbage classification, we found it necessary to present children with all the system information related to garbage classification, so they can understand how garbage will be treated after classification and what effect it will produce. By directly linking the results with the classification behavior, children can better understand the necessity of garbage classification. An engaging, interactive format and personified images attract children’s attention, enhance their desire to use the toy, and result in stronger educational outcomes.

Based on the above strategies, we designed an intelligent trash can system toy. A contrast experiment was carried out to verify the effect of our design strategies. After playing with the intelligent trash can system toy for a period of time, the children displayed a 31.25% improvement in their garbage classification accuracy. At the same time, this intervention training promoted children’s garbage-sorting behavior in daily life. When they saw others misclassifying trash, they would correct these mistakes and take the initiative to share relevant knowledge about garbage disposal.

This study provides detailed design strategies for preschool garbage-sorting toy products, which provide a good reference for designers and enterprises for design practice and manufacturing. It provides a new idea for improving the effects of garbage classification education for preschool children. Presenting children with all the system information related to garbage classification can help them better understand the necessity of garbage classification. This study also proves that interactive-experience toys are a feasible means of providing garbage-sorting education for preschool children. Using these types of toys will improve children’s garbage-sorting behavior and awareness. In order to achieve the best educational effects, our design ideas and educational forms should keep pace with the times and conform to the current children’s cognition and interests. This research is an innovative attempt to design intelligent interactive toys. The combination of technology and design innovation can not only bring us comfortable and convenient life experience, but also promote us to have a green lifestyle.

## Figures and Tables

**Figure 1 ijerph-20-04460-f001:**
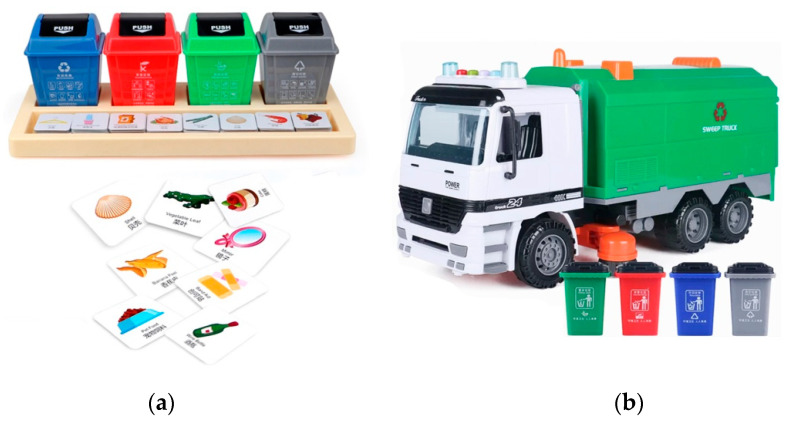
(**a**) Card game and (**b**) garbage truck toy.

**Figure 2 ijerph-20-04460-f002:**
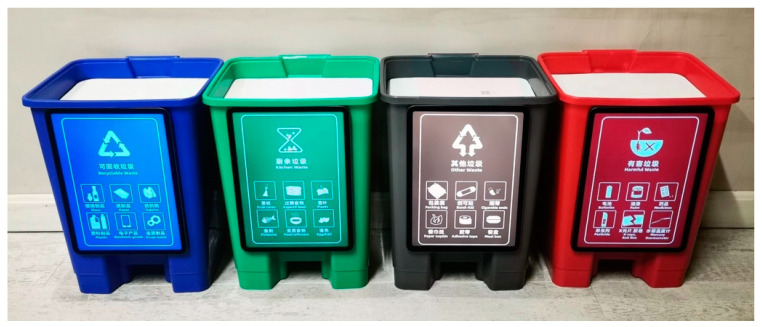
The designed intelligent trash can system toy.

**Figure 3 ijerph-20-04460-f003:**
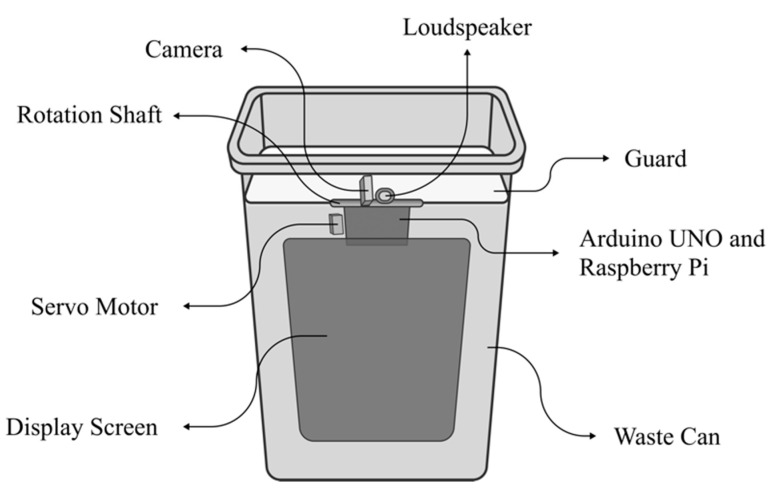
Components of the designed intelligent trash can.

**Figure 4 ijerph-20-04460-f004:**
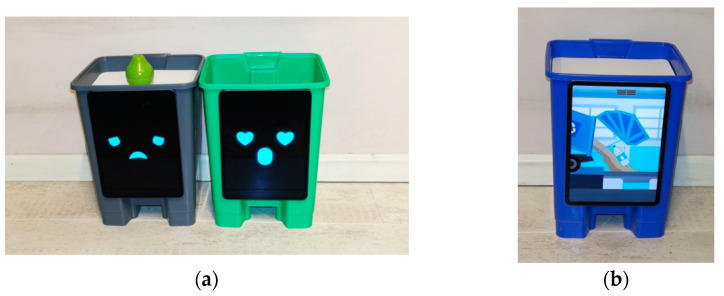
Expressive reactions of the designed intelligent trash can system toy. The toy displayed different expressions to distinguish whether users threw the garbage into the wrong trash can or not (**a**). The garbage thrown in the trash was displayed in the form of cute cartoons on the monitor (**b**).

**Figure 5 ijerph-20-04460-f005:**
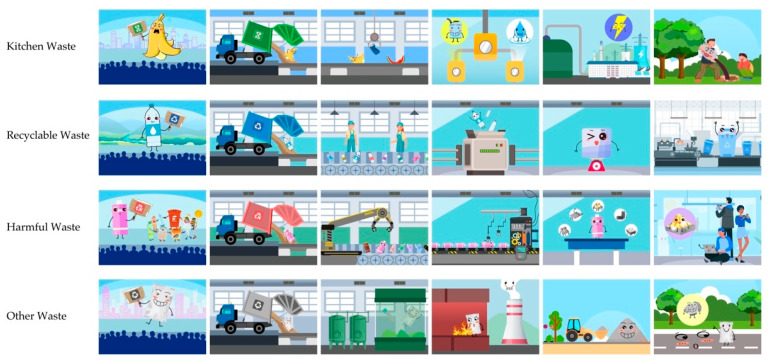
Key animation frames of the representative disposal of each type of waste.

**Figure 6 ijerph-20-04460-f006:**
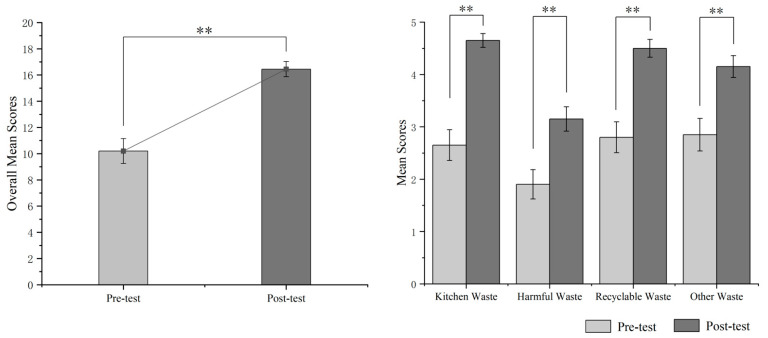
Comparison of delivery accuracy between pre-test and post-test. ** *p* < 0.01.

**Figure 7 ijerph-20-04460-f007:**
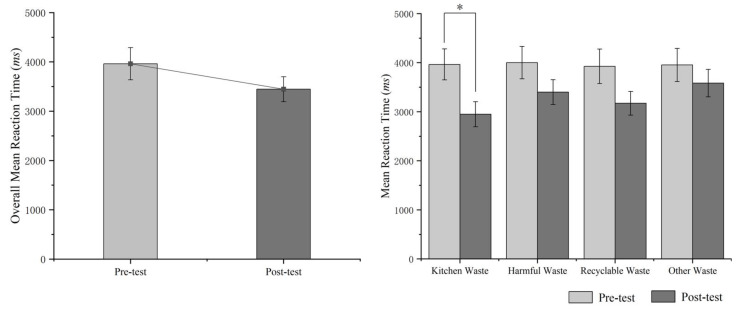
Comparison of reaction duration between pre-test and post-test. * *p* < 0.05.

**Table 1 ijerph-20-04460-t001:** Type and function comparison of preschool garbage classification educational toys and preschool educational toys.

	Preschool Garbage Classification Educational Toys	Preschool Educational Toys
**Type**	1. Cards 52%	1. Lego toys 36%
	2. Garbage truck toys 44%	2. Chess and card games 10%
		3. Tool-type toys 34%
	3. Waste-sorting board games 4%	4. Life simulation toys 14%
		5. Remote-control toys 6%
**Function**	1. Practice correct garbage classification	1. Systematic knowledge learning
	2. Understand the garbage truck workflow	2. Skill learning
		3. Intellectual development

**Table 2 ijerph-20-04460-t002:** Summary of design strategies.

**Educational Content**	Systematic knowledge	1. Garbage classification knowledge
		2. How to dispose of garbage
		3. Effects after garbage treatment
**Use**	Interactive, timely feedback	
**Form**	Interesting, attractive information via images	
**Atmosphere**	Relaxed, happy	

## Data Availability

Link to publicly archived datasets analyzed during the study: https://osf.io/3ptgd/ (accessed on 30 December 2022).

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
