# Peer review of "Interactive Educational Toy Design Strategies for Promoting Young Children’s Garbage-Sorting Behavior and Awareness"

_ijerph, 2023, doi:10.3390/ijerph20054460_

Round 1
Reviewer 1 Report
This paper aims to improve young children’s garbage-sorting behavior and strategies. To do so, the authors start with a summary of the best-selling garbage-classification educational toys. Then, an interview was delivered to parents of 20 children after they experienced two most common preschool garbage-classification educational toys, followed by a summary of design strategies were provided. Based on this, the authors proposed their educational system and conducted a user study with 20 children to evaluate the system. The results showed that the proposed system could help improve the memorization of garbage classification with high satisfaction.
This is an interesting project and addresses an important topic in today's society that consumes a large quantity of goods. Engaging children in recycling issues, especially in places where it is not widely practiced, via a creative and interactive tool is an important way to influence their behavior early.
The following are the strengths of the paper.
+ Hot and valuable topic.
+ Thorough research on the related toys in the market.
+ Thorough analysis of the study results, which are interesting and useful.
Below are some areas that could be clarified further:
- Additional information can be provided in Section 2.1. For example, many readers outside of China may not be familiar with Taobao. Also, there was mention of “impact assessment” but was not defined.
- Similarly, a few more details could be included in Section 2.2. For example, how were the toys delivered? Did the children receive both toys at the beginning? Or they had each toy per week?
- Finally, were the participants in the user study the same as in the previous interview described in Section 2.2?
Reviewer 2 Report
Dear Authors,
The manuscript presents a study that explores the impact of educational-toy design strategies on promoting young children’s garbage-sorting behavior and awareness. The study presents interesting findings that highlight the effectiveness of certain toy design strategies in promoting children’s garbage-sorting behavior. The manuscript is well-written and provides a comprehensive overview of the research.
General Comments:
The study is well-designed, and the methodology is clearly presented. The results are significant, and the implications of the research are clearly drawn. However, there are a few minor changes that the authors should make to improve the manuscript.
Major Comments:
Previous Literature: The authors should include a section on the previous literature on the topic of educational-toy design strategies for promoting young children’s garbage-sorting behavior and awareness. This would help readers to better understand the research and its contributions to the existing body of knowledge.
Gender Classification of Results: The results section could be improved by adding the gender classification of results. This would be more significant for future recommendations.
Conclusion: The conclusion section is a repetition of the results. The authors should add some fine outcomes and a few bullet points to show the contribution of research. This would help to strengthen the conclusion and highlight the importance of the research.
Minor Comments:
The title could be more descriptive to reflect the focus of the study.
The authors should provide more details about the study population.
The authors should add more details about the statistical analysis conducted.
Recommendation:
Overall, the manuscript presents a well-designed study with significant results. The minor changes suggested above would improve the manuscript and make it more impactful. I recommend that the manuscript be accepted with minor revisions.
Thank You
